# Smartwatch-based detection of loss of pulse

**Berken Utku Demirel**[1]*, **Galip Utku Akay**[2], **Paul Streli**[1],
**Hatice Ozturkmen**[3], **Christian Holz**[1]

**1** ETH Zurich, Zurich, Switzerland, **2** Université University of Caen Normandie, Caen, France, **3** Baskent University, Istanbul, Turkiye

* berkenutku.demirel@gmail.com

## Abstract

We present a framework and a novel dataset for detecting loss of pulse cases using a smartwatch. Our methodology for gathering data involves inducing controlled pulse loss in the forearm within a clinical setting under medical supervision. To the best of our knowledge, this is the first multimodal dataset comprising photoplethysmogram (PPG) signals, inertial measurements (IMUs) that can be obtained from a commercially available smartwatch, and continuous recordings as ground truth, collected from 20 individuals. This dataset serves as a valuable resource for the research community to advance and verify techniques for detecting pulse loss and differentiate between the non-usage of smartwatches to prevent false alarms. Our presented framework consists of extracting several features from signals and employing machine learning models designed to differentiate between the emergency loss of pulse cases with the non-usage of smartwatches. The presented framework achieves a detection accuracy of up to 97.1%, with a false detection rate of 1.0 in an hour when evaluated using leave-one-subject-out cross-validation.

## 1 Introduction

Early detection and timely intervention are critical for survival in out-of-hospital pulse loss cases [1,2], but delays often occur due to the absence of continuous real-time cardiovascular health monitoring techniques [3,4]. Detecting loss of pulse is particularly crucial in unwitnessed cases, which constitute the majority of these incidents [5]. In such situations, individuals are unable to seek help or initiate intervention, similar to the case with out-of-hospital cardiac arrest patients [6,7]. The absence of immediate medical attention in these scenarios drastically reduces the chances of survival [8], making real-time, automated detection systems essential for initiating timely intervention to improve the survival ratio.

   Detecting the loss of pulse through wearable devices, such as smartwatches, represents an advancement in the effort to enhance early detection and response time while improving the continuous monitoring of individuals outside of clinical environments [4,9,10]. Although several commercially available smartwatches detect falls and alert emergency services, no devices can concomitantly assess the presence of

**Data availability statement:** All relevant data are available at the following link: https://github.com/eth-siplab/Loss_of_Pulse.

**Funding:** The author(s) received no specific funding for this work.

**Competing interests:** The authors have declared that no competing interests exist.

a pulse [1]. Recently, Google introduced the "Pixel Watch 3," which features novel loss of pulse detection [11]. This feature represents the first integration of machine learning techniques for automated pulse loss detection using a commercial smartwatch outside of a hospital setting.

However, the research community lacks datasets with ground truth measurements to improve techniques for detecting loss of pulse using smartwatches. Moreover, despite the introduction of new smartwatch features for detecting pulse loss, the sensitivity and specificity of the deployed algorithms remain unverified [1], which can increase unnecessary emergency calls due to false positives. For example, in recent years, many smartwatches have introduced fall-detection features that have been instrumental in saving lives [12]. However, these features have also led to false positives, resulting in unnecessary emergency calls and potentially diverting critical resources away from other urgent situations [13]. Thus, a publicly available dataset using a commercial device is crucial for researchers to refine and enhance algorithms for detecting pulse loss, enabling more accurate and reliable performance in real-world scenarios.

Collecting data on loss of pulse is inherently challenging, particularly in settings with commercial smartwatches. One of the primary difficulties is the extremely short duration of pulse loss before medical intervention occurs, making it difficult to gather large, continuous datasets. This limited availability of data poses a significant challenge for training data-hungry machine learning models, especially deep learning algorithms that typically require large volumes of data to achieve high performance. In most cases, loss of pulse events are rare and often quickly addressed, further limiting the opportunity to collect comprehensive datasets. This scarcity makes it difficult to capture the wide variability in physiological signals associated with pulse loss, which is crucial for developing robust detection algorithms.

Considering these limitations, in this work, we introduce a publicly available multimodal dataset for detecting pulse loss using a commercial smartwatch. To ensure the dataset contains a continuous and sufficiently large sample size for reliable evaluation, data was collected during a medical assessment, where blood flow to the upper extremity was restricted using a cuff under medical supervision. Along with this dataset, we develop a machine learning-based framework designed to detect pulse loss while minimizing the false positive rate. Our dataset, collected from 20 individuals, includes PPG and IMU signals from a commercially available smartwatch for detecting pulse loss. Additionally, we collected continuous Doppler Ultrasonography (US) imaging as ground truth to confirm the absence of blood circulation in forearm.

## 1.1 Contributions

The contributions of this paper are as follows:

- A framework for detecting the loss of pulse detection algorithm that considers the PPG and inertial measurements together to increase the sensitivity while decreasing the false positive ratio to prevent unnecessary emergency alerts.

- An extensive evaluation of our proposed framework using a novel dataset while including sensitivity, specificity, and ablation analyses to investigate the contribution of each component.
- A multimodal dataset that includes PPG, inertial measurements, and continuous ultrasound recordings as ground truth, collected from 20 individuals. To the best of our knowledge, our dataset is the first multimodal dataset for detecting pulse loss with a commercial smartwatch.

## 2 Related works

Detecting vital signs and physiological signals using wearable devices, particularly smartwatches, has been an area of growing interest [14–16]. Various studies have explored using PPG and IMU signals for monitoring heart rate [17–19], and activity, and even detecting critical events like falls and arrhythmias [13,20]. While extensive literature has demonstrated the potential of commercial smartwatches for ambulatory health monitoring and emergency detection [21], the specific challenge of detecting pulse loss—crucial in cases such as cardiac arrest [3,6]—has only recently begun to receive attention in the research community [1]. For example, a recent work induced short-lasting circulatory arrests as part of routine practice (transcatheter aortic valve implantation, defibrillation testing, or ventricular tachycardia induction) to detect these events with a PPG wristband during the procedure [6].

While the recent studies on detecting circulatory arrests using a PPG wristband present a novel approach, they have several limitations. First, the focus of the experiments was exclusively on circulatory arrest events, which narrows the scope of its findings. Moreover, the population sample was limited to patients with pre-existing medical conditions, excluding individuals without a medical record. This lack of a control group and broader demographic restricts the generalizability of methods, as the findings may not fully represent the wider population or be applicable across diverse settings. Given our focus on designing a framework for use in smartwatches aimed at the general population, it is important to evaluate the method on individuals without a medical history to ensure its broad applicability.

Additionally, most of the previous studies did not incorporate inertial measurement units as an additional modality. The only usage of PPG is a drawback, as IMUs provide valuable information about the movements, which could enhance the accuracy and contextual understanding of the data from the smartwatch. Furthermore, the loss of pulse durations was induced during major operations such as transcatheter aortic valve implantation or defibrillation testing [1]. Patients undergoing these procedures are under medication and anesthesia, conditions that significantly affect vessels and blood flow [22,23]. This procedural context conditions the effects on vessels differently [23], potentially influencing the outcome. Additionally, this experimental setup limits both the sample size and the duration of segments with loss of pulse as they were induced during surgical operations, thereby restricting the study's ability to provide a comprehensive analysis of circulatory arrests over varied conditions and timeframes.

In a recent development, Google introduced the "Pixel Watch 3", featuring a loss of pulse detection system detailed by Shah et al. [24]. Their work demonstrates the viability of using a commercial smartwatch for this purpose, employing a multi-stage pipeline that processes PPG and motion data through a lightweight convolutional neural network trained on 528 engineered features. Notably, Shah et al. utilized arterial occlusion to simulate pulse loss, validating the relevance of the occlusion methodology used in our own study. This approach is crucial for realism, as it allows for data collection from healthy, conscious participants, avoiding the confounding physiological effects of anesthesia and medication inherent to intra-operative clinical studies.

However, despite these advancements, the underlying datasets remain private, and ground truth is often established via secondary indicators like standard pulse oximetry. Our work addresses these gaps by providing a multimodal dataset. Unlike standard clinical labeling, our use of continuous ultrasound imaging also enables us to visually confirm and precisely annotate the exact millisecond of blood flow cessation in the radial artery, minimizing label noise and ensuring the model is trained on high-fidelity physiological endpoints.

Another major limitation is that none of these efforts have released their datasets for public use to further advance the field by developing new techniques. Some works have achieved specificity as high as 99.9%, but since the datasets are private and the developed methods rely on multiple threshold values optimized specific to datasets, it remains unclear whether these approaches will generalize to other conditions. This lack of accessible data has hindered further advancements in the field, particularly in developing and refining methods for detecting loss of pulse in commercial smartwatches. Furthermore, the absence of standardized benchmarks and evaluation protocols makes it difficult to compare methods objectively and identify the most promising approaches. Consequently, further research is essential to develop and validate methods in diverse, real-world scenarios for this critical application.

In contrast to previous studies, our work is unique in both its focus on pulse loss detection using a smartwatch with PPG and inertial measurements and its commitment to transparency while contributing a publicly available multimodal dataset to the community for advancing the research in the area. To overcome the limitations of sample size and realistic induction of loss of pulse, we introduced a novel methodology while using a commercially available smartwatch.

## 3 Methodology

### 3.1 Study design and participants

Our study design is focused on healthy adults (aged 18 years or older) in whom there is no medical record of cardiovascular abnormalities, such as arrhythmias, heart failure, or coronary artery disease. Additional exclusion criteria were set as having medical issues interfering with the wearing of the smartwatch. The study was conducted at Baskent University in Turkey between August 2023, and August 2024, and was set up by a research consortium consisting of the Baskent University faculty of medicine, Istanbul Uygulama araştırma Hastanesi, Turkey), and ETH Zurich (Zurich, Switzerland). Consecutive patients who met the inclusion criteria were selected for the study. The study protocol was approved with the reference number KA23/123 by the Medical Research Ethics Committee of Baskent University. Written informed consent was obtained from all participants before inclusion.

### 3.2 Data collection

Study participants were equipped with a Samsung Smartwatch device (Samsung Gear S3 Frontier) during the data collection. PPG data with multi-wavelength—paired green, red, and infrared sensors—is recorded with a sampling frequency of 128 Hz during the entire procedure and sent to a computer.

As a gold standard, Doppler US, (Canon Aplio I800 TUS- A1800, Tokyo Japan) data were continuously collected by 11 L4 PLT-704 SBT linear array transducer and stored during the study. The Doppler US data primarily in- vestigates the radial artery.

The data collection started with subjects seated comfortably to ensure a relaxed state. Before the start of the procedure, an adult-size sphygmomanometric blood pressure cuff (Welch Allyn cuff, New York, United States) was placed right side to 2–3 cm above the antecubital fossa, with the cuff size adjusted according to the subject's arm circumference. Blood pressure was measured and recorded. And then the subjects wore the smartwatch on the right wrist. We make sure that the usage of the smartwatch is similar to daily usage, therefore, the device was worn on the wrist according to the subject's preference with the preferred tightness level. Doppler US was placed right next to smartwatch. The radial artery was imaged on axial plane using Doppler US. While Doppler US and smartwatch remained in this location, both of them simultaneously recorded data throughout the experiment. An example setup of the data collection is shown in Fig 1. The experiment consisted of three main stages: Stage 1: It was recorded a two-minute baseline to capture initial conditions. Stage 2: A blood pressure cuff, placed on the subject's right forearm, was inflated to 50 mmHg above systolic pressure, occluding blood flow in the radial artery. The occlusion was shown with Doppler US at the same time. The blood flow was obstructed for maximum five minutes to collect enough data for training. In the final stage, the clinician released the pressure, and the recording continued for an additional three minutes to observe the vessel flow returned to baseline levels

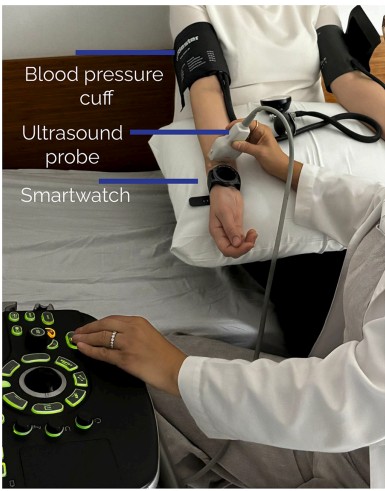

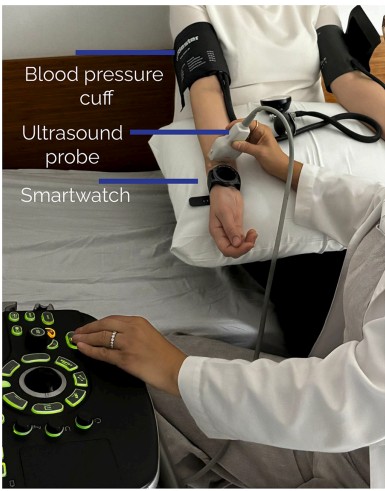

**Fig 1**. The data collection setup with the positions of people and the clinician for recording PPG data with ground truth Doppler US signal.

All critical points during the procedure were annotated by clinicians on the Doppler US screen for data analysis. Baseline variables, including age, body mass index, and sex were also collected. Overall, we collected 20.35 hours of smartwatch data from 20 subjects (11 males, 9 females) with a mean age of 32.7 years, of which approximately 1.40 hours correspond to the loss of pulse events. The detailed duration for each event is provided in Table 1. We also recorded PPG and inertial measurements from the smartwatch when it was not being worn by the users. Differentiating between these periods is crucial, as the framework must distinguish between the absence of a pulse and a lack of signal due to the smartwatch not being worn, thereby reducing false positives.

### 3.3 Event definition

In this study, we detected the exact timing of the loss of pulse events using the recorded Doppler US data by a trained clinician. The detection process primarily relies on Doppler US, which provides real-time information about blood flow [25]. When the Doppler US indicates that blood flow has reached zero, it signifies the loss of pulse. At this critical point, the clinician annotates the event in Doppler US to ensure precise analysis. Examples of the event definition for the baseline (2a), the pulse loss (2b and 3a), and the return of blood flow (3a) cases are given in Figs 2 and 3. To maintain labeling precision, we excluded transition segments——those occurring during the onset or offset of pulse loss—from our classification. This was done to avoid ambiguity and ensure the model is trained and evaluated on clearly defined pulsating and pulseless states. To ensure accurate alignment between data from different devices, we synchronized the smartwatch recordings with the Doppler US data, achieving a time difference of less than 10 ms. This ground truth method allows for accurate and reliable detection of pulse loss compared to previous approaches, which often relied on secondary indicators. For instance, in some studies, circulatory arrests were defined using rapid ventricular pacing during aortic

**Table 1**. Summary of duration and segment count by case.

| Case | Duration (hours) | # of segments |
|---|---|---|
| Loss of Pulse | 1.40 | 2517 |
| Usage of Smartwatch | 14.97 | 27537 |
| Non-Usage of Smartwatch | 3.98 | 7161 |
| **Total** | 20.35 | 37215 |

A

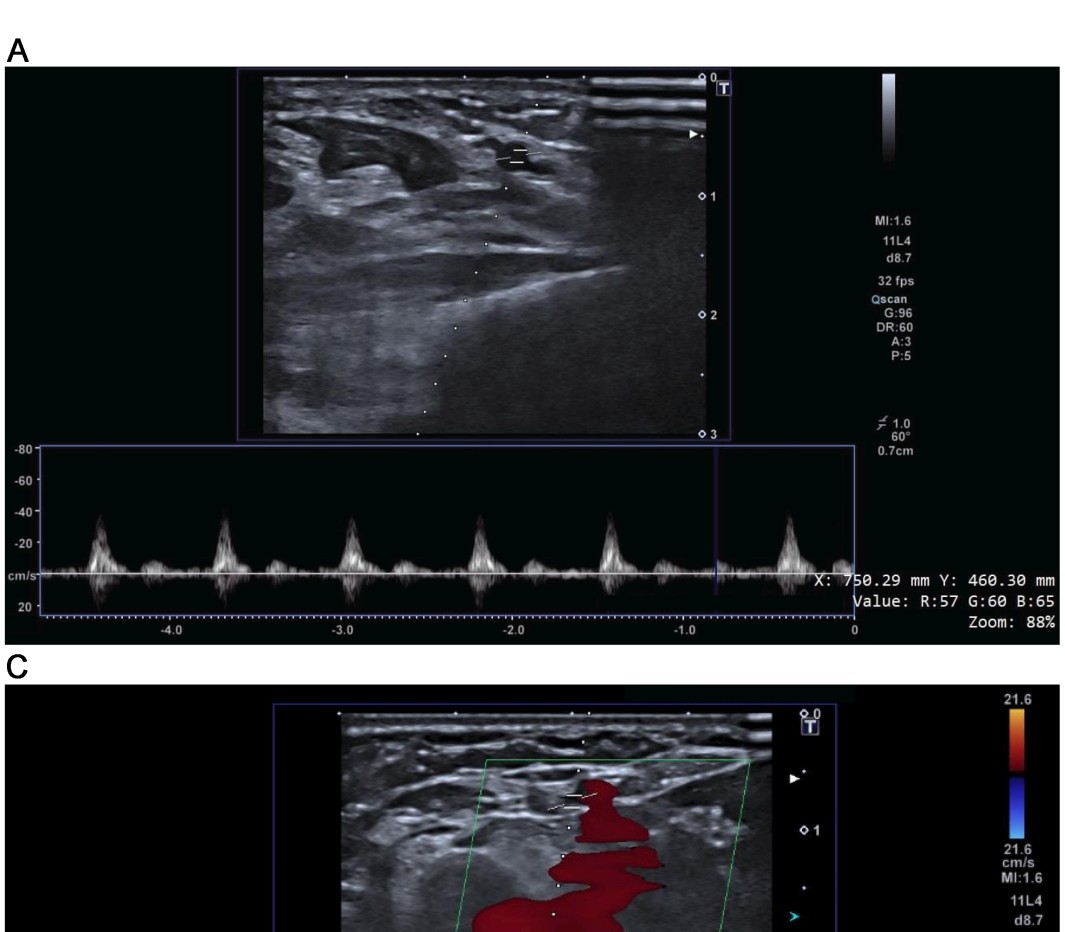

C

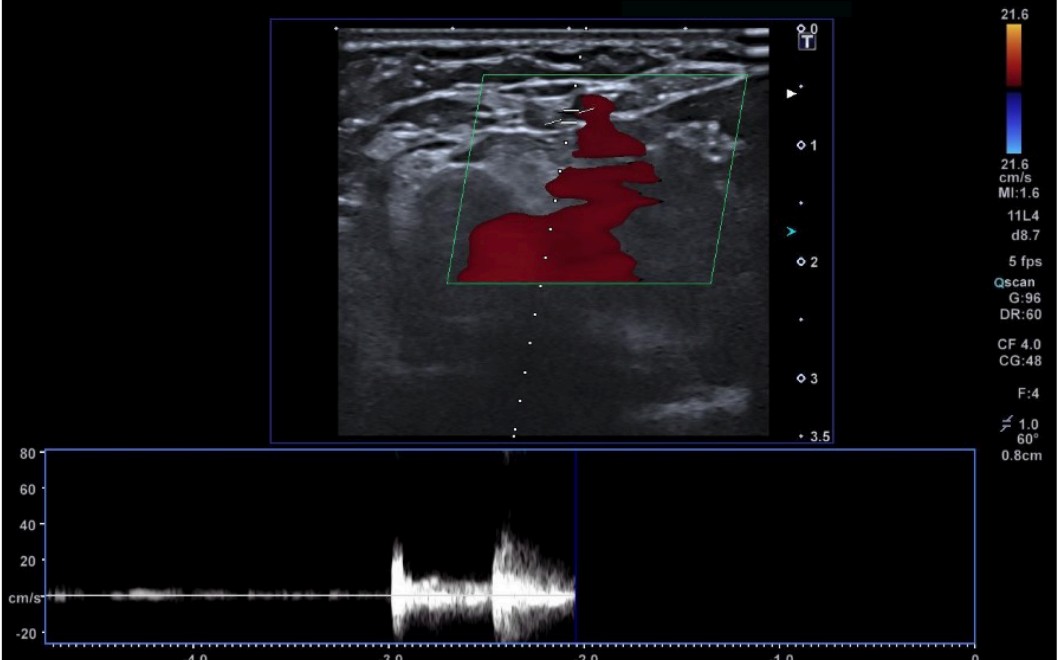

Fig 2. **Doppler US showing the event annotations.** At the beginning of the recording, there is no blood flow to the forearm until the Doppler US shows it. (a) A Doppler US snapshot of the baseline recording (no intervention), showing normal blood flow with heartbeats). (b) A Doppler US for detecting the exact timing for the start of the blood flow after loss of pulse.

balloon inflation [1], resulting in a short-term circulatory standstill. Although this procedure is routinely performed during balloon-expandable valve placement and balloon dilation, either before or after valve implantation, in cases where balloon dilation was not performed, the detection of pulse loss could be less precise. Our use of Doppler US to directly measure blood flow and annotate the exact moment of pulse loss by a clinician provides a more robust and dependable method for event detection.

**A**

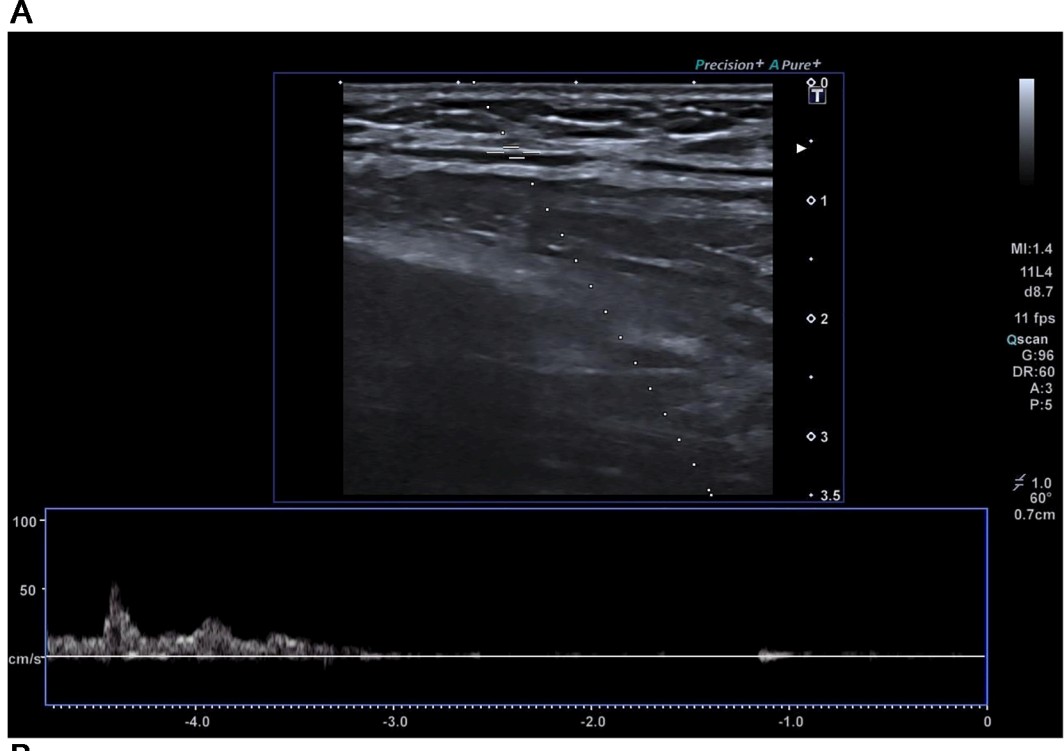

**B**

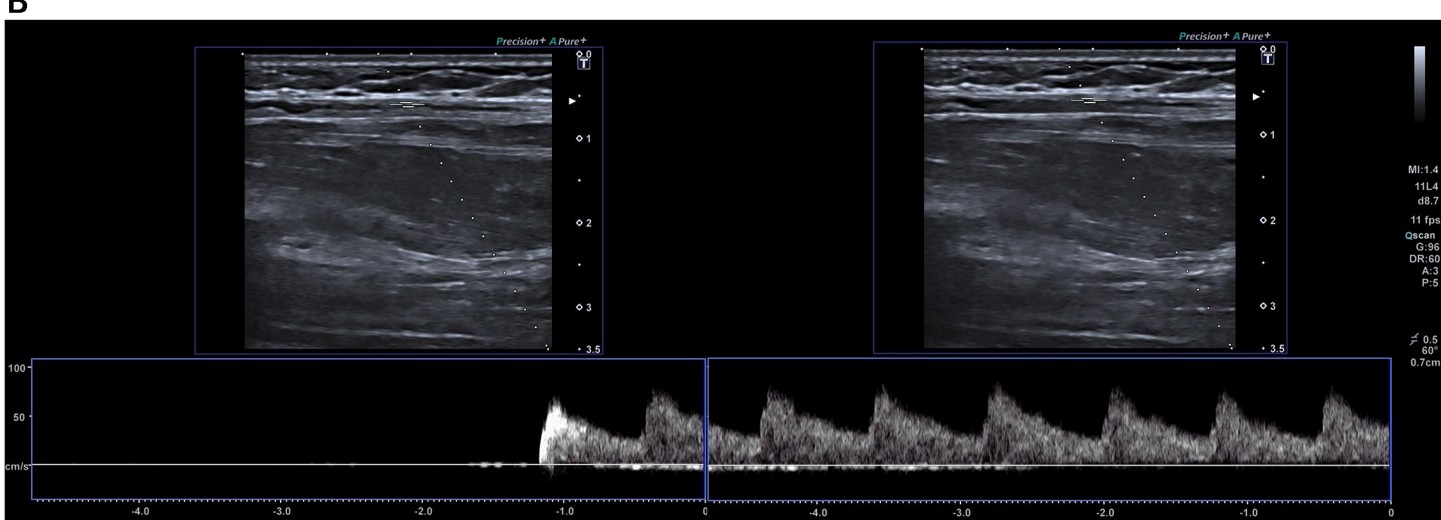

**Fig 3**. **Doppler ultrasound snapshots illustrating the two critical events in our protocol: (a) loss of pulse following cuff inflation, indicated by the cessation of blood flow, and (b) restoration of pulse after cuff deflation, marked by the reappearance of blood flow.** (a) A Doppler US snapshot showing loss of pulse after cuff pressure is applied. As shown, blood flow drops to zero following occlusion. (b) Extended Doppler ultrasound showing the return of blood flow after cuff release, included to clearly depict pulse restoration.

### 3.4 Processing

Our processing framework starts with a filtering step to standardize and denoise the collected data. For the PPG signals, first, we apply a fourth-order Butterworth bandpass filter with 0.5–25 Hz. Next, we resample the signal to 100 Hz while applying an 8-second sliding window with 2-second shifts for segmentation. Then, we averaged three channels together

to form a single channel PPG. During our pre-processing pipeline, we have not applied any normalization to the signals as the variance of the signals give information about the blood flow.

For the inertial measurements, we used the raw signals from the 3-axis accelerometers, segmented in the same manner as the PPG signals without applying any filtering or normalization.

### 3.5 Event classification

We employed two approaches to classify events into three categories and compared their performance. In both approaches, the events were classified as pulse loss, normal operation (the subject wearing the smartwatch without intervention), and idle (when the smartwatch is not worn by any subject). The details of each framework are given below.

**3.5.1 Deep learning based framework.** First, we used a deep learning method where the processed signals are fed as inputs to an architecture where the output of the architecture is one of the three events. Specifically, we design a neural network architecture that includes one encoder for each modality: one for PPG and one for inertial measurements. Each encoder is used to extract modality-specific features from the signals independently before concatenating the features. Encoders consist of a convolutional layer of 16 kernels of the size of 2 and an LSTM cell with a hidden size of 32. We apply batch normalization [26] after each convolutional layer. We concatenate the features extracted from the two encoders and pass them through three fully connected linear layers with sizes of 10, 20, and 40, respectively. We use the ReLU activation function between linear layers, with a sigmoid activation applied at the end to classify the signal segments into one of the three classes. The overall architecture has approximately 500k parameters. We also give a schematic illustration of our architecture in Fig 4.

**Training architecture.** The designed architecture takes as input only the pre-processed PPG and IMU segments and no other patient-related features. We used the Adam optimizer [27] with $\beta_1 = 0.9$, $\beta_2 = 0.999$, and a mini-batch size of 32. The learning rate was initialized to $5e{-}4$ with a weight decay of $1e{-}6$ and reduced by 0.1 when the validation loss stopped improving for 10 consecutive epochs. The training continues until 30 successive epochs without validation performance improvements. The best model is chosen as the lowest specificity on the validation data.

We implemented signal processing parts (filtering, resampling) in MATLAB (MathWorks, USA) [28] and Python, depending on the format of the publicly available dataset. The proposed dual-stream CNN-LSTM architecture was implemented in PyTorch [29]. The experiments are carried out on an NVIDIA GeForce RTX 3090 GPU.

**3.5.2 Feature extracting based framework.** In this work, we propose a feature extraction-based framework to classify the loss of pulse using PPG and IMU signals from a commercially available smartwatch. The framework includes data preprocessing, feature extraction, and classification using traditional machine learning methods.

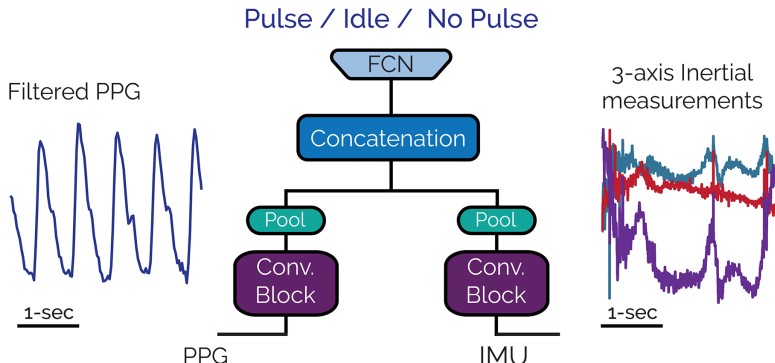

**Fig 4**. **The used architecture for neural network based loss of pulse detection based on two modalities for three classess.**

We extract various time-domain, frequency-domain, and morphological features from the PPG and IMU signals. Features were extracted independently from each channel, one PPG and three IMU axes, and then concatenated as input to the classifiers. Overall, we used ten features from the PPG channel and 21 features from the IMU signals (7 features per axis × 3 axes). Although we explored the performance of the models using additional features [30], such as wavelets [31], we did not observe any significant improvement in the metrics. Table 2 lists the features used in this study.

After obtaining the features from signals, we investigated several traditional classifiers based on machine learning, including:

- Binary Decision Trees [32] with a leaf size of 4 and 36, respectively, to compare the impact of the small and big trees on the performance.
- Simple linear regressions [33] to explore the effectiveness of our features using a simple model structure.
- Support vector machines [34] with linear and Gaussian kernels, respectively, to examine the effect of more generalized ML structures.
- Gaussian process (GPR) [35] with the exponential and squared exponential as the kernel functions in covariance matrices, respectively.
- Ensemble trees [36] with Boosted and Bagged settings, respectively, to observe the impact of ensemble modeling approaches that use multiple regression trees jointly.

## 3.6 Evaluation

We evaluate the performance of the classifiers using leave-one-subject-out cross-validation, with performance metrics including sensitivity, specificity, and false detection per hour (FD/h). To evaluate the model's capability in generalizing to unseen users, we used leave-one-subject-out (LOSO) cross-validation where the model is trained using all other users' data and evaluated on a specific user. This is also beneficial in real scenarios, as our model does not require data recorded from users before deploying the model (i.e., no user-specific training is performed).

$$\text{Sensitivity (Se)} = \frac{TP}{TP + FN} \tag{1}$$

**Table 2**. **Extracted features from PPG and IMU signals.**

| Feature | Domain |
| --- | --- |
| **PPG Features** | |
| Mean | Time |
| Standard Deviation | Time |
| Skewness | Time |
| Kurtosis | Time |
| RMSSD | Time |
| Power in Frequency Bands | Frequency |
| LF/HF Ratio | Frequency |
| Spectral Entropy | Frequency |
| Pulse Maximum Amplitude | Morphological |
| Pulse Width | Morphological |
| **IMU Features** | |
| Mean Acceleration | Time |
| Standard Deviation | Time |
| Energy | Time |
| Tilt Angle | Time |
| Jerk (Mean, Max) | Time |
| Dominant Frequency | Frequency |
| Spectral Entropy (IMU) | Frequency |

$$\text{Specificity (Sp)} = \frac{TN}{TN + FP} \qquad (2)$$

$$FD/h = \frac{FP}{\text{Total Recording Time in Hours}}, \qquad (3)$$

True Positives (TP) refer to the correct identification of pulse loss events, while True Negatives (TN) represent the accurate classification of normal or non-usage smartwatch cases. False Negatives (FN) occur when the model incorrectly classifies a pulse loss event as normal or non-usage. Finally, False Positives (FP) occur when the model incorrectly identifies a pulse loss event, despite the actual case being normal or non-usage of the smartwatch. Although the model is trained for three-class classification, we report sensitivity and specificity by combining the normal and non-usage classes into a single category (label 0), treating pulse loss as the positive class (label 1). This reflects our primary goal of accurately detecting pulse loss events. To provide a more complete evaluation across all three classes, we also report the macro-averaged F1 score.

## 4 Experimental results and analysis

We evaluated the performance of our presented framework with fixed architecture and hyperparameters across subjects. Table 3 gives the metrics computed in our proposed dataset following the scheme explained in Sect 3.

Table 3 illustrates the detailed performance of the frameworks, the employed dual-stream CNN-LSTM that takes PPG and IMU signals as input, and the feature extraction-based classifiers.

Our results indicate that non-linear models generally outperform their linear counterparts. Specifically, the support vector machines with Gaussian kernels and the Gaussian process models show significant improvements over the simpler linear models. The Gaussian Process with the squared exponential kernel achieved the sensitivity of 89.7% and the lowest false detection rate at 1.9 FD/h.

Ensemble methods, including both Boosted and Bagged trees, also demonstrated superior performance compared to linear regression. The Boosted Ensemble Trees achieved the highest sensitivity of 90.6% and a false detection rate of 1.7 FD/h, demonstrating their ability to handle non-linear relationships effectively.

Among all methods evaluated, the employed dual-stream CNN-LSTM model gave the best overall performance in three metrics with a sensitivity of 92.8%, specificity of 97.1%, and the lowest false detection rate of 1.0% FD/h. Although the performance difference between the traditional learning based approachs and the dual-stream CNN-LSTM is similar, our results underscore the capability of deep learning approaches to model intricate non-linear relationships in the data. This suggests that the extracted features have complex interactions with the target classes, which traditional methods struggle

**Table 3**. **Comparison of model performance with combined PPG and IMU signals.**

| Method | Se (%) | Sp (%) | FD/h | F1 (%) |
|---|---|---|---|---|
| Binary Decision Trees (Leaf 4) | 84.8 | 91.7 | 2.6 | 88.0 |
| Binary Decision Trees (Leaf 36) | 87.4 | 93.6 | 2.1 | 90.0 |
| Linear Regression | 81.2 | 88.4 | 4.0 | 84.5 |
| SVM (Linear Kernel) | 86.7 | 93.2 | 2.3 | 89.5 |
| SVM (Gaussian Kernel) | 88.6 | 94.5 | 1.9 | 91.0 |
| Gaussian Process (Exponential) | 86.1 | 92.3 | 2.6 | 89.2 |
| Gaussian Process (Squared Exp.) | 89.7 | 94.3 | 1.9 | 91.5 |
| Ensemble Trees (Boosted) | 90.6 | 95.7 | 1.7 | 92.7 |
| Ensemble Trees (Bagged) | 88.3 | 93.6 | 2.1 | 90.9 |
| Dual-stream CNN-LSTM (Ours) | **92.8** | **97.1** | **1.0** | **94.8** |

to fully exploit—especially evident in the confusion between the non-usage of the smartwatch and actual pulse loss. Furthermore, this observation implies that the current feature set may not be optimal for this task, and further efforts in feature extraction and engineering could enhance performance across all methods.

These findings highlight the importance of employing non-linear models for pulse loss detection. The superior performance of the deep learning model further emphasizes the information in raw data for leveraging complex relationships.

Fig 5 presents representative examples of PPG signals corresponding to various classification outcomes, including true positives, false positives, true negatives, and false negatives. These examples illustrate both accurate and erroneous classifications, offering insight into the model's behavior across different event types.

From this figure, we observed that in the absence of a true pulse, the PPG signal often resembles white Gaussian noise. When such noise is passed through filters with narrow cut-off frequencies, it can take on a pseudo-periodic appearance that mimics a normal pulse pattern. This effect can lead to misclassifications by the model.

These findings suggest that signal normalization and filtering steps should be applied with caution in loss-of-pulse detection tasks. In particular, normalization techniques should consider the power and structure of the input signal to avoid introducing misleading patterns that may reduce model reliability.

## 4.1 Ablation study

In this section, we investigated the different components of the overall framework and their individual contributions to the performance. Our focus was mainly on exploring the contribution of sensor modalities to the creation of the dataset. To assess the effectiveness of these components, we kept the parameters of the learning models unchanged throughout the experiments. We also perform the same evaluation scheme described in Sect 3.6.

**4.1.1 Sensor modalities.** To assess the impact of different sensor modalities on the performance of the pulse loss detection framework, we conducted an ablation study where we systematically varied the number of modalities used in the classification process and investigate the performance in three metrics. Since our main experiment employs both PPG and IMU data combined, in our ablation study, we separately evaluated the performance using PPG-only and IMU-only data to understand the contribution of each modality and the benefits of their combination.

Table 4 summarizes the results of the ablation study, with each row representing a different classification model and its performance metrics for a specific modality. For the dual-stream CNN-LSTM model, we excluded a specific branch corresponding to the modality under evaluation. For the other models, features related to the excluded modality were omitted during both training and inference to ensure consistency in the comparison.

From the results, we can observe that using both PPG and IMU modalities together yields the highest sensitivity (92.8%) and specificity (97.1%) with the lowest false detection rate (1.0 FD/h). This demonstrates the complementary nature of PPG and IMU data in improving the accuracy and reliability of the framework for the detection of pulse loss.

When evaluating the PPG-only case, sensitivity decreased to 92.1% and the false detection rate increased to 1.3 FD/h, indicating that PPG data alone is less effective in capturing the full context of pulse loss compared to the combined modality. Similarly, using IMU-only data resulted in the lowest sensitivity of 89.1% and the highest false detection rate of 1.8 FD/h, underscoring the insufficiency of IMU data alone for reliable performance.

A notable finding from our ablation experiments is that IMU-only methods caused a more significant performance decline compared to PPG-only methods, highlighting the relatively higher contribution of PPG data in detecting pulse loss. This suggests that while both modalities are essential for optimal performance, PPG data plays a more critical role, and the IMU data acts as a complementary signal to refine the detection process.

Overall, the results from the ablation study confirm the importance of using both PPG and IMU modalities together for comprehensive pulse loss detection. The combined use of these modalities not only improves the sensitivity and

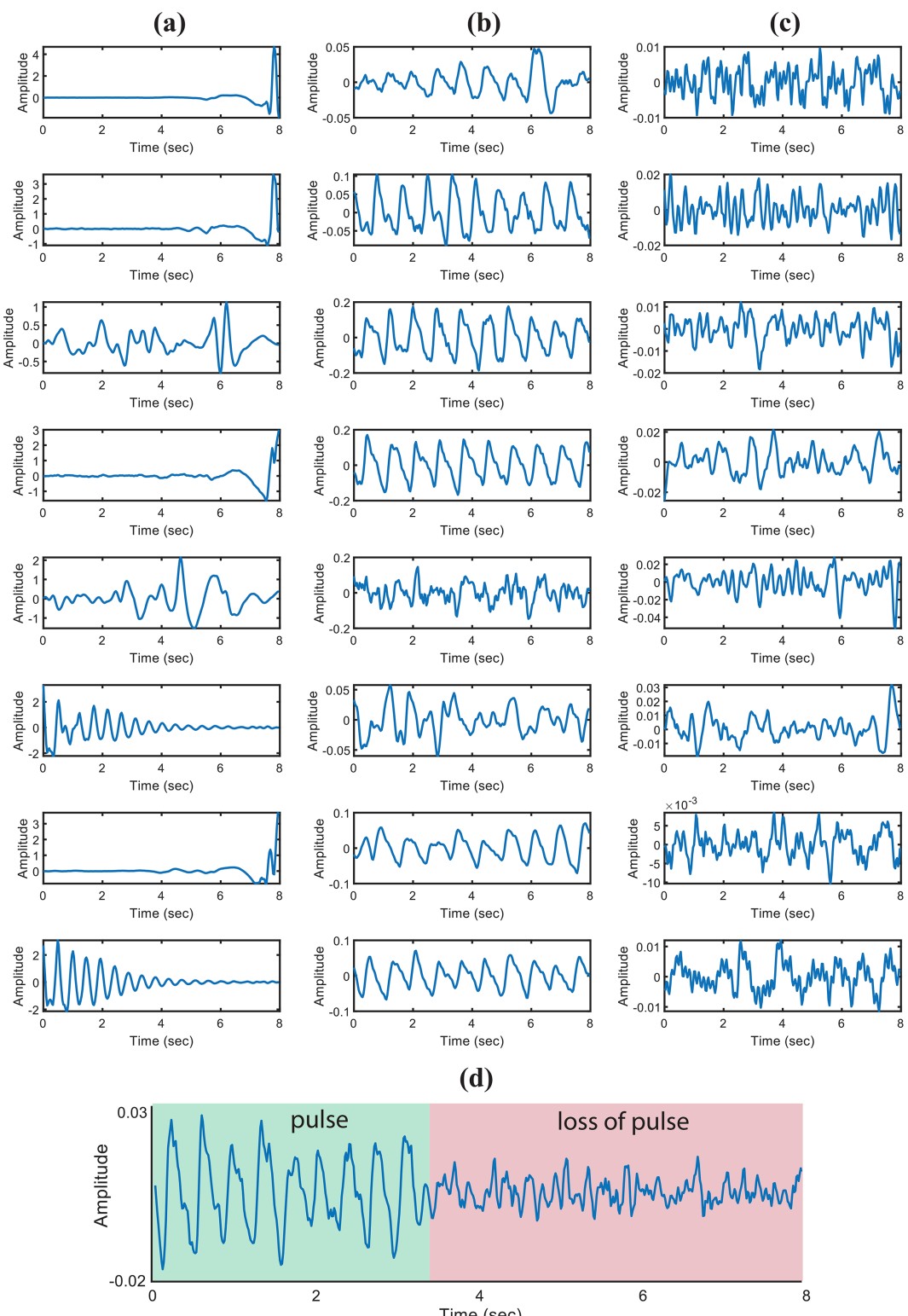

**Fig 5**. **Representative (PPG) signals illustrating various detection outcomes: (a) false negatives, (b) true positives, (c) false positives, and (d) a case demonstrating the transition from a normal pulse to pulse loss.**

**Table 4.** Performance of models using PPG and IMU modalities separately.

| (a) Performance using only PPG | | | | |
|---|---|---|---|---|
| **Method** | **Se (%)** | **Sp (%)** | **FD/h** | **F1 (%)** |
| Binary Decision Trees (Leaf 4) | 84.1 | 90.6 | 2.8 | 87.0 |
| Binary Decision Trees (Leaf 36) | 86.3 | 92.8 | 2.4 | 89.5 |
| Simple Linear Regression | 80.4 | 87.6 | 4.3 | 83.5 |
| SVM (Linear Kernel) | 86.1 | 92.7 | 2.6 | 89.0 |
| SVM (Gaussian Kernel) | 88.2 | 93.6 | 2.1 | 90.5 |
| Gaussian Process (Exponential) | 85.4 | 91.6 | 2.8 | 88.0 |
| Gaussian Process (Squared Exp.) | 89.2 | 93.5 | 2.2 | 90.9 |
| Ensemble Trees (Boosted) | 89.6 | 95.1 | 2.0 | 92.0 |
| Ensemble Trees (Bagged) | 86.8 | 93.2 | 2.5 | 89.8 |
| Dual-stream CNN-LSTM (Ours) | 92.1 | 95.0 | 1.3 | **93.5** |
| (b) Performance using only IMU | | | | |
| **Method** | **Se (%)** | **Sp (%)** | **FD/h** | **F1 (%)** |
| Binary Decision Trees (Leaf 4) | 81.9 | 88.3 | 3.6 | 85.0 |
| Binary Decision Trees (Leaf 36) | 83.2 | 89.0 | 2.7 | 86.0 |
| Simple Linear Regression | 77.5 | 85.9 | 4.8 | 81.0 |
| SVM (Linear Kernel) | 83.6 | 90.2 | 3.5 | 87.0 |
| SVM (Gaussian Kernel) | 86.1 | 90.8 | 3.1 | 88.5 |
| Gaussian Process (Exponential) | 81.6 | 87.6 | 3.5 | 84.5 |
| Gaussian Process (Squared Exp.) | 85.3 | 90.5 | 2.7 | 87.9 |
| Ensemble Trees (Boosted) | 88.1 | 92.3 | 2.6 | 90.2 |
| Ensemble Trees (Bagged) | 84.7 | 90.7 | 2.5 | 87.7 |
| Dual-stream CNN-LSTM (Ours) | 89.1 | 93.4 | 1.8 | **91.5** |

specificity but also reduces the number of false detections, thereby enhancing the overall performance of the detection framework.

**4.1.2 Excluding training data.** We performed an additional ablation experiment where the models were trained without using the data segments of a non-usage smartwatch case. However, during the evaluation phase, all the events were included to simulate a more realistic scenario where the model might encounter data from both pulse loss and smartwatch non-usage. The results of this experiment are presented in Table 5. As seen in the table, excluding non-usage data during training caused a significant drop in performance across all metrics. Specifically, sensitivity and specificity decreased considerably, while the false detection rate increased substantially. This performance drop was observed across all methods; however, for simplicity, we report detailed comparisons only for the best-performing dual-stream CNN-LSTM model. Notably, the performance decline in traditional methods was even worse, further emphasizing the importance of including non-usage smartwatch data in training to ensure robustness in real-world scenarios where the smartwatch may not always be worn.

As the results suggest, incorporating all relevant classes, including the non-usage case, during training is critical to achieving high classification accuracy. This ablation further emphasizes the challenge of generalizing to unseen data when specific scenarios are not accounted for during training.

**4.1.3 Performance in-the-real world data.** To evaluate the practicality of our approach in real-life conditions, we tested our algorithm on the publicly available DaLiA dataset [37], which contains wrist-worn PPG and IMU data collected

**Table 5.** Impact of excluding non-usage data during training on pulse loss detection performance.

| Training cases | Se (%) | Sp (%) | FD/h |
|---|---|---|---|
| PPG + IMU (Main Experiment) | 92.8% | 97.1% | 1.0 |
| Excluding Non-Usage Data (Ablation) | 76.3% | 80.6% | 4.4 |

from 15 subjects during various daily activities such as walking, sitting, climbing stairs, and cycling. The dataset includes approximately 3 hours of recordings per subject, resulting in a diverse and challenging real-world dataset with natural variations in heart rate, motion, and signal quality.  When applied to the DaLiA dataset, our method achieved a lower false detection rate ranging between 0.3 and 0.5 per hour. This improvement is notable compared to the 1.0 false detections per hour observed in our more controlled experiments, particularly those where the smartwatch was not worn. These results indicate that our approach generalizes well to real-world scenarios where the watch is worn naturally and users exhibit dynamic, realistic physiological responses.

## 5 Discussion and future work

In this work, we presented a framework and a novel multimodal dataset for detecting the loss of pulse using a commercial smartwatch. Our approach focuses on addressing a critical gap in the literature, specifically detecting pulse loss, which is an essential but less explored area in the era of wearables. The presented frameworks which are a deep learning architecture and extracting hand-crafted features from signals are designed to differentiate between the loss of pulse case with the usage/non-usage of smartwatches. The best classifier achieved a detection accuracy of up to 97%, with a false detection rate of 0.7 in an hour when evaluated using leave-one-subject-out cross-validation.

However, despite these promising results, questions remain about whether the proposed approach's performance is excellent. Therefore, it is important to evaluate the limitations of our work. First, in this work, since we focus on detecting loss of pulse, we did not simulate fall events in this study. The primary reason is that numerous datasets and well-established techniques already exist for fall detection [38,39], and our goal was to focus on the lacking component in the area.

By concentrating solely on the loss of pulse, we aim to provide a framework for further research. However, we acknowledge that integrating fall detection and pulse loss events could offer a more comprehensive solution for real-world scenarios in which a fall might trigger a subsequent loss of pulse. Thus, further research and effort should be undertaken to collect more comprehensive datasets that combine fall detection with pulse loss, allowing for the exploration of end-to-end techniques that encompass both events. Such a dataset would likely facilitate more advanced, ideally explainable [40], models capable of handling the entire sequence of events, improving real-time emergency detection capabilities in wearable technologies.

Second, subjects wore the smartwatch in their preferred arms within a specific position to observe the blood vessels using the ultrasound device. This procedure can be more relaxed in future works where the subjects also position the smartwatch as they wish for a more realistic experiment using a smaller transducer.

Third, in our experiments, we have not tried to extract heartbeats from inertial measurements using the ballistiocardiogram (BCG). Extracting heartbeats using BCG is common and widely used in health applications [41–44], especially during periods when there is less motion [45]. Thus, BCG signals can also be used to detect if there is any loss of pulse while decreasing the false detection rate.  However, in our setup, we did not observe clear BCG patterns in the smartwatch's IMU signals. This may be due to a combination of motion noise and the limited sensitivity of the smartwatch's inertial sensors, which might not be sufficient to capture the subtle mechanical vibrations associated with cardiac activity. We have therefore excluded BCG-based features from our current analysis. Nonetheless, we acknowledge the potential of BCG for pulse detection and encourage future work to explore its utility using higher-sensitivity sensors or under more controlled conditions.

Fourth, regarding the state-of-the-art comparison, we note that while Shah et al. [24] also utilized arterial occlusion to simulate pulse loss, their detection pipeline relies on proprietary, device-specific firmware logic. Specifically, the authors use dynamic adjustment of LED currents and photodiode gain upon initial detection to verify pulselessness. Because our dataset consists of passively recorded signals from a standard consumer smartwatch without such active hardware control, a direct application of their specific multi-stage algorithm to our data was not feasible. Future work could address this

by developing hardware-agnostic approximations of these active sensing protocols, or by using programmable research platforms to facilitate direct benchmarking across different devices.

Finally, we acknowledge that using a cuff to simulate pulse loss does not fully replicate the physiological conditions of true loss of pulse, which might involve arrhythmias and complex cardiovascular dynamics. While the cuff method is a widely accepted approach in controlled settings, and has been used in similar prior studies to simulate sudden loss of perfusion [24], it remains a proxy. Therefore, we would like to note that although our method allows for reproducible data collection and model validation, its clinical fidelity to actual loss of pulse is limited.

Our work provides a foundation for further exploration into detecting the loss of pulse with wearable devices. By addressing a key gap in the field, we encourage future studies to expand on this work by integrating fall detection, incorporating signals like BCG, and developing more comprehensive public datasets. These efforts could lead to more effective and practical solutions for real-world health monitoring applications.

## 6 Conclusion

In this paper, we introduce a framework and novel multimodal dataset for detecting the loss of pulse using a commercial smartwatch. By leveraging PPG signals, inertial measurements, and continuous ultrasound recordings, our dataset provides an important resource for advancing research in continuous health monitoring using wearable technology. The framework demonstrated strong performance, achieving up to 97.1% detection accuracy with a low false detection rate in three realistic scenarios. We believe that these results highlight the potential of using smartwatches for early and reliable detection of critical health events regarding loss of pulse, paving the way for future applications and techniques in continuous health monitoring.

## Author contributions

**Conceptualization:** Berken Utku Demirel, Hatice Ozturkmen, Christian Holz.

**Data curation:** Galip Utku Akay, Hatice Ozturkmen.

**Funding acquisition:** Christian Holz.

**Methodology:** Berken Utku Demirel.

**Project administration:** Christian Holz.

**Resources:** Christian Holz.

**Software:** Paul Streli.

**Supervision:** Christian Holz.

**Writing – original draft:** Berken Utku Demirel.

**Writing – review & editing:** Berken Utku Demirel, Hatice Ozturkmen, Christian Holz.

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
