## [Decision Letter · Decision Letter 0]

22 Jul 2025

PONE-D-25-29278Smartwatch-Based Detection of Loss of PulsePLOS ONE

Dear Dr. Demirel,

Thank you for submitting your manuscript to PLOS ONE. After careful consideration, we feel that it has merit but does not fully meet PLOS ONE’s publication criteria as it currently stands. Therefore, we invite you to submit a revised version of the manuscript that addresses the points raised during the review process.

Two reviewers have provided their feedback below. Please address each point in turn. In particular, the reviewers ask for clarification in a number of areas and request that you check for consistent use of specific terminology.

We look forward to receiving your revised manuscript.

Kind regards,

Joanna Tindall, PhD

Staff Editor

PLOS ONE

Journal Requirements:

3. Please include a complete copy of PLOS’ questionnaire on inclusivity in global research in your revised manuscript. Our policy for research in this area aims to improve transparency in the reporting of research performed outside of researchers’ own country or community. The policy applies to researchers who have travelled to a different country to conduct research, research with Indigenous populations or their lands, and research on cultural artefacts. The questionnaire can also be requested at the journal’s discretion for any other submissions, even if these conditions are not met. Please find more information on the policy and a link to download a blank copy of the questionnaire here: https://journals.plos.org/plosone/s/best-practices-in-research-reporting. Please upload a completed version of your questionnaire as Supporting Information when you resubmit your manuscript.

Reviewers' comments:

Reviewer's Responses to Questions

**Comments to the Author**

1. Is the manuscript technically sound, and do the data support the conclusions?

Reviewer #1: Yes

Reviewer #2: Yes

2. Has the statistical analysis been performed appropriately and rigorously?

Reviewer #1: Yes

Reviewer #2: Yes

3. Have the authors made all data underlying the findings in their manuscript fully available?

Reviewer #1: No

Reviewer #2: No

4. Is the manuscript presented in an intelligible fashion and written in standard English?

Reviewer #1: Yes

Reviewer #2: Yes

5. Review Comments to the Author

Reviewer #1: Thank you for this timely and important contribution. Your work introduces a novel dataset and framework for smartwatch-based detection of pulse loss. However, I have several important concerns and questions regarding the clarity and completeness of the methods, evaluation, and claims that need to be addressed.

1. Classification Labels and Evaluation Metrics

The paper states that the task is a three-class classification problem: pulse loss, normal operation, and idle (non-usage of smartwatch). However, Figure 4 and several parts of the text seem to only reference binary classification (e.g., "pulse" vs. "no pulse").

The metrics presented (sensitivity and specificity) are defined and reported in the binary classification context. Please clarify:

Are you performing binary or multiclass classification?

If multiclass, please provide per-class sensitivity and specificity, or macro- or weighted-averaged values for all classes.

Figure 4 and its caption should clearly show all 3 classes, if that’s the intention.

2. Terminology: "Deep Learning" vs. "Neural Network"

Throughout the manuscript, you refer to your deep learning model as a "neural network" and at times suggest it's an autoencoder. The current architecture more closely resembles a dual-encoder fusion model with LSTM layers, not an autoencoder.

Please revise the terminology to reflect the actual structure (e.g., “dual-stream CNN-LSTM” or “multi-modal encoder network”) to avoid confusion.

3. Proposed Method Identification

It is unclear which model is considered your proposed method: is it the deep learning architecture or one of the ensemble ML classifiers?

Please explicitly label the "proposed model" in the tables and main text and clearly distinguish between baseline comparisons and the final model.

4. Feature Dimensions and Signal Processing

You extracted features from all 3 IMU channels. Please clarify:

What is the final shape of the input feature vector to the classifiers?

Did you compute features for each axis individually and then concatenate them?

Or did you average features across the three axes before feeding to the model?

You state that no filtering or normalization was applied to IMU signals. Please justify this, especially since IMU signals tend to be noisy and scale-dependent.

Did you apply any resampling or time-alignment between PPG and IMU signals?

5. Ambiguity in "IMU" Definition

Please clearly define what you mean by “IMU” in the context of your device:

Are you using accelerometer only, gyroscope only, or both?

If both are present, are they all used in the signal fusion?

6. Cross-Validation and Subject Separation

You mention using Leave-One-Subject-Out (LOSO) cross-validation. Please confirm:

Was this strict LOSO, with no subject data leakage during feature extraction or normalization?

Was LOSO also applied to the deep learning model, or just traditional classifiers?

7. PPG-DaLiA Evaluation and Ground Truth

You evaluate your model on the PPG-DaLiA dataset, which does not contain ground truth for pulse loss or smartwatch idle status.

How did you define the labels for this dataset?

How did you differentiate normal vs. pulse loss vs. non-usage in the absence of ground truth?

Please explain how Doppler Ultrasound, used as ground truth in your dataset, relates to the evaluation on PPG-DaLiA, where no such ground truth is available.

8. Comparison with Existing Methods

The manuscript does not provide any benchmark comparison with previous methods in pulse detection or loss-of-pulse detection.

Please compare your model with at least one existing baseline or state-of-the-art method to contextualize the reported accuracy.

Reviewer #2: The paper presents a system based on smartwatches with the aim to detect the loss of pulse. The paper is well writen and structured although some improvements can be achieved:

- The smartwatch (Samsung Gear S3 Frontier) characteristics should be listed as well as the characteristics of the smartwatch sensors (vendor, resolution, maximum range...)

- The sentence "An example setup of the data collection is shown in Figure 1" is writen twice in page 4

- The reference to Figure 3 in page 5 is not clear. I am not sure if it is really to Figure 3 or Figure 2. Check it. Consequently, Figure 2 or 3 are not referenced or explained in the text as only one is referenced.

- The paper compares the performance of some models. However they are not explained in detail. That is, the parameters of each model are not presented. For instance, what is the architecture of the employed Neural Network Model? The obtained performance using a NN is very dependant of its architecture.

- Table 4 (in page 11) should be placed after (or closer) to its reference (page 13).

- The sentence in page 15 "Second, when the subjects wore ..." seems to be incomplete.

6. PLOS authors have the option to publish the peer review history of their article (what does this mean?). If published, this will include your full peer review and any attached files.

Reviewer #1: **Yes:** Kianoosh Kazemi

Reviewer #2: **Yes:** Francisco J. González-Cañete

---

## [Author Response · Author response to Decision Letter 1]

5 Aug 2025

Thank you for the constructive feedback. Due to character limitations, we have provided detailed point-by-point responses to all reviewer comments in the attached PDF (appended at the end). We kindly ask the reviewers to refer to the attachment for the full responses and clarifications.

---

## [Decision Letter · Decision Letter 1]

23 Oct 2025

PONE-D-25-29278R1Smartwatch-Based Detection of Loss of PulsePLOS ONE

Dear Dr. Demirel,

Thank you for submitting your manuscript to PLOS ONE. After careful consideration, we feel that it has merit but does not fully meet PLOS ONE’s publication criteria as it currently stands. Therefore, we invite you to submit a revised version of the manuscript that addresses the points raised during the review process.

We look forward to receiving your revised manuscript.

Kind regards,

Diaa Ahmed Mohamed Ahmedien, Ph.D.

Academic Editor

PLOS ONE

Journal Requirements:

**Additional Editor Comments:**

Dear authors,

Thank you for your careful revision, it would be helpful to further consider the first revierwer's comments as a final revision.

Best regards

Reviewers' comments:

Reviewer's Responses to Questions

**Comments to the Author**

1. If the authors have adequately addressed your comments raised in a previous round of review and you feel that this manuscript is now acceptable for publication, you may indicate that here to bypass the “Comments to the Author” section, enter your conflict of interest statement in the “Confidential to Editor” section, and submit your "Accept" recommendation.

Reviewer #1: (No Response)

Reviewer #2: All comments have been addressed

2. Is the manuscript technically sound, and do the data support the conclusions?

Reviewer #1: Yes

Reviewer #2: Yes

3. Has the statistical analysis been performed appropriately and rigorously?

Reviewer #1: Yes

Reviewer #2: N/A

4. Have the authors made all data underlying the findings in their manuscript fully available?

Reviewer #1: No

Reviewer #2: No

5. Is the manuscript presented in an intelligible fashion and written in standard English?

Reviewer #1: Yes

Reviewer #2: Yes

6. Review Comments to the Author

Reviewer #1: I would encourage the authors to discuss and compare their approach with other recent work on smartwatch-based loss of pulse detection. For instance, Shah et al. (NShah, K., Wang, A., Chen, Y. et al. Automated loss of pulse detection on a consumer smartwatch. Nature 642, 174–181 (2025). https://doi.org/10.1038/s41586-025-08810-9) presented an automated loss of pulse detection system implemented on a consumer smartwatch, using a multi-stage pipeline with PPG, motion data, and a lightweight convolutional neural network trained on 528 engineered features.

Given that your work also focuses on smartwatch-based pulse loss detection with machine learning, it would strengthen the manuscript to position your framework in relation to this paper, highlighting methodological similarities (e.g., multimodal PPG + motion sensing, simulation of pulse loss via occlusion) as well as differences (e.g., dataset openness, model architectures, evaluation settings).

Reviewer #2: The requirements of the reviewers seem to be accomplished, at least those made by me in the first round of review.

For future responses to reviewers I suggest to include the full text of the reviewer, instead of a short sentence, in order to be understandable by the rest of reviewers or even to be remembered by the reviewer that is doing the second round of reviews. That way, the reviewers know exactly what was requested in the first round of reviews.

7. PLOS authors have the option to publish the peer review history of their article (what does this mean?). If published, this will include your full peer review and any attached files.

Reviewer #1: **Yes:** Kianoosh Kazemi

Reviewer #2: No

---

## [Author Response · Author response to Decision Letter 2]

22 Nov 2025

We would like to thank the reviewers for their useful comments on our paper. We have provided point-by-point responses to their comments and questions. We have also made several changes to the paper in response to their concerns.

In the updated manuscript, all the changes are marked in Blue.

The replies to the itemized comments are included where the responses are colored Green, and the corresponding changes to the manuscript are pasted and colored Blue.

In addition to the uploaded PDF, we have included the reviewers’ comments and our point-by-point responses below.

Reviewer 1 comments/questions:

I would encourage the authors to discuss and compare their approach with other recent work on smartwatch-based loss of pulse detection. For instance, Shah et al. (NShah, K., Wang, A., Chen, Y. et al. Automated loss of pulse detection on a consumer smartwatch. Nature 642, 174–181 (2025). https://doi.org/10.1038/s41586-025-08810-9) presented an automated loss of pulse detection system implemented on a consumer smartwatch, using a multi-stage pipeline with PPG, motion data, and a lightweight convolutional neural network trained on 528 engineered features.

Given that your work also focuses on smartwatch-based pulse loss detection with machine learning, it would strengthen the manuscript to position your framework in relation to this paper, highlighting methodological similarities (e.g., multimodal PPG + motion sensing, simulation of pulse loss via occlusion) as well as differences (e.g., dataset openness, model architectures, evaluation settings).

Response:

We thank the reviewer for the suggestion.

We agree that positioning our work in relation to Shah et al. (Nature, 2025) strengthens the manuscript.

We have updated Section 2 (Related Work) to acknowledge their use of arterial occlusion while highlighting our unique contributions. Furthermore, we have added a paragraph to Section 5 (Discussion) to clarify why a direct algorithmic comparison was not feasible due to the proprietary, hardware-specific nature of their approach.

Changes to Section 2: Related Work

In a recent development, Google introduced the "Pixel Watch 3", featuring a loss of pulse detection system detailed by Shah et al.

Their work demonstrates the viability of using a commercial smartwatch for this purpose, employing a multi-stage pipeline that processes PPG and motion data through a lightweight convolutional neural network trained on 528 engineered features.

Notably, Shah et al. utilized arterial occlusion to simulate pulse loss, validating the relevance of the occlusion methodology used in our own study.

This approach is crucial for realism, as it allows for data collection from healthy, conscious participants, avoiding the confounding physiological effects of anesthesia and medication inherent to intra-operative clinical studies.

However, despite these advancements, the underlying datasets remain private, and ground truth is often established via secondary indicators like standard pulse oximetry.

Our work addresses these gaps by providing a multimodal dataset.

Unlike standard clinical labeling, our use of continuous ultrasound imaging also enables us to visually confirm and precisely annotate the exact millisecond of blood flow cessation in the radial artery, minimizing label noise and ensuring the model is trained on high-fidelity physiological endpoints.

Changes to Section 5: Discussion

Fourth, regarding the state-of-the-art comparison, we note that while Shah et al also utilized arterial occlusion to simulate pulse loss, their detection pipeline relies on proprietary, device-specific firmware logic.

Specifically, the authors use dynamic adjustment of LED currents and photodiode gain upon initial detection to verify pulselessness.

Because our dataset consists of passively recorded signals from a standard consumer smartwatch without such active hardware control, a direct application of their specific multi-stage algorithm to our data was not feasible.

Future work could address this by developing hardware-agnostic approximations of these active sensing protocols, or by using programmable research platforms to facilitate direct benchmarking across different devices.

---

## [Decision Letter · Decision Letter 2]

19 Jan 2026

Smartwatch-Based Detection of Loss of Pulse

PONE-D-25-29278R2

Dear Dr. Demirel,

We’re pleased to inform you that your manuscript has been judged scientifically suitable for publication and will be formally accepted for publication once it meets all outstanding technical requirements.

Kind regards,

Agnese Sbrollini

Academic Editor

PLOS One

Reviewers' comments:

Reviewer's Responses to Questions

**Comments to the Author**

1. If the authors have adequately addressed your comments raised in a previous round of review and you feel that this manuscript is now acceptable for publication, you may indicate that here to bypass the “Comments to the Author” section, enter your conflict of interest statement in the “Confidential to Editor” section, and submit your "Accept" recommendation.

Reviewer #1: All comments have been addressed

Reviewer #2: All comments have been addressed

2. Is the manuscript technically sound, and do the data support the conclusions?

Reviewer #1: Yes

Reviewer #2: Yes

3. Has the statistical analysis been performed appropriately and rigorously?

Reviewer #1: Yes

Reviewer #2: Yes

4. Have the authors made all data underlying the findings in their manuscript fully available?

Reviewer #1: No

Reviewer #2: Yes

5. Is the manuscript presented in an intelligible fashion and written in standard English?

Reviewer #1: Yes

Reviewer #2: Yes

6. Review Comments to the Author

Reviewer #1: The manuscript is written scientifically. The authors has addressed all the comments. and the discussion is now fair with comparing their method with other existing methods.

Reviewer #2: The requirements of the reviewers seem to be accomplished. Hence, it can be accepted as is without further modificaciones.

7. PLOS authors have the option to publish the peer review history of their article (what does this mean?). If published, this will include your full peer review and any attached files.

Reviewer #1: **Yes:** KIANOOSH KAZEMI

Reviewer #2: **Yes:** Francisco Javier González Cañete

---

## [Editor Report · Acceptance letter]

PONE-D-25-29278R2

PLOS One

Dear Dr. Demirel,

I'm pleased to inform you that your manuscript has been deemed suitable for publication in PLOS One. Congratulations! Your manuscript is now being handed over to our production team.

Kind regards,

on behalf of

Dr. Agnese Sbrollini

Academic Editor

PLOS One